# Design and Accuracy of an Instrumented Insole Using Pressure Sensors for Step Count

**DOI:** 10.3390/s19050984

**Published:** 2019-02-26

**Authors:** Armelle M. Ngueleu, Andréanne K. Blanchette, Laurent Bouyer, Désirée Maltais, Bradford J. McFadyen, Hélène Moffet, Charles S. Batcho

**Affiliations:** 1Centre for Interdisciplinary Research in Rehabilitation and Social Integration (CIRRIS), Centre intégré universitaire de santé et de services sociaux de la Capitale-Nationale (CIUSSS-CN), Quebec City, QC G1M2X8, Canada; armelle.ngueleu.1@ulaval.ca (A.M.N.); andreanne.blanchette@fmed.ulaval.ca (A.K.B.); Laurent.Bouyer@rea.ulaval.ca (L.B.); Desiree.Maltais@rea.ulaval.ca (D.M.); Brad.McFadyen@fmed.ulaval.ca (B.J.M.); Helene.Moffet@rea.ulaval.ca (H.M.); 2Department of Rehabilitation, Faculty of Medicine, Université Laval, Quebec City, QC G1M2X8, Canada

**Keywords:** insole, pressure sensors, algorithms, step count, accuracy, healthy participant

## Abstract

Despite the accessibility of several step count measurement systems, count accuracy in real environments remains a major challenge. Microelectromechanical systems and pressure sensors seem to present a potential solution for step count accuracy. The purpose of this study was to equip an insole with pressure sensors and to test a novel and potentially more accurate method of detecting steps. *Methods*: Five force-sensitive resistors (FSR) were integrated under the heel, the first, third, and fifth metatarsal heads and the great toe. This system was tested with twelve healthy participants at self-selected and maximal walking speeds in indoor and outdoor settings. Step counts were computed based on previously reported calculation methods, individual and averaged FSR-signals, and a new method: cumulative sum of all FSR-signals. These data were compared to a direct visual step count for accuracy analysis. *Results*: This system accurately detected steps with success rates ranging from 95.5 ± 3.5% to 98.5 ± 2.1% (indoor) and from 96.5 ± 3.9% to 98.0 ± 2.3% (outdoor) for self-selected walking speeds and from 98.1 ± 2.7% to 99.0 ± 0.7% (indoor) and 97.0 ± 6.2% to 99.4 ± 0.7% (outdoor) for maximal walking speeds. Cumulative sum of pressure signals during the stance phase showed high step detection accuracy (99.5 ± 0.7%–99.6 ± 0.4%) and appeared to be a valid method of step counting. *Conclusions*: The accuracy of step counts varied according to the calculation methods, with cumulative sum-based method being highly accurate.

## 1. Introduction

Advances in technology during the past decades have led to circuit boards being increasingly miniaturized, which has facilitated the design of wearable measuring and recording devices. These devices have application in the health care system, in particular in rehabilitation for monitoring, and training [1,2,3]. Some wearable devices such as pedometers, accelerometers, activity monitors are used to measure and record daily step counts [4,5,6,7,8,9,10] which are an important indicator of energy expenditure in daily life activity [11]. Pedometers and accelerometers are commercially available low-cost devices [5,7,9,12,13,14]. However, studies have shown that, when used individually, pedometers and accelerometers demonstrate low accuracy (0.19 ≤ Intraclass Correlation Coefficient (ICC) ≤ 0.73) in detecting steps in stroke survivors and in people with traumatic brain injury [15,16]. In contrast, activity monitors that include accelerometer, gyroscope and magnetometer sensors [17,18,19] have been shown to accurately reflect the number of steps taken with more than 90% of accuracy in the same populations [15,16]. In older adults, current knowledge related to the accuracy of the activity monitors in step counting remains limited [20]. A study by Hergenroeder et al. evaluated seven activity monitors (Fitbit Charge, Garmin Vivofit, Fitbit Zip, Yamax SW-200 Digiwalker, Accusplit Pedometer, Omron Pedometer, Yamax EX-510 Pedometer) in elderly participants and reported accuracies from 93.68 ± 13.95% to 39.1 ± 40.3% depending on walking speed [20]. Studies have reported no significant difference of step count for Fitbit Charge HR and observer counted (criterion) in young people [21] and for Fitbit Zip and an ActiGraph GT3x-BT accelerometer in children [22]. In another study, accuracies were high (0.78 ≤ ICC ≤ 0.98) for StepWatch, Fitbit One and ActivPAL and low (0.12 ≤ ICC ≤ 0.4) for Fitbit Charge, Fitbit One, G-Sensor, Garmin Vivofit, Actigraphin compared with observed step count in young people [23]. In younger participants, one study reported a step count error that varied from 41.3 ± 13.8% to 0.04 ± 4.3% for various activity monitors (Fitbit One, Actigraph, Jawbone, Fitbit Charge, Fitbit Flex) [24].

Another potential and promising approach for the identification of step count may be to integrate pressure sensors into shoe insoles [11,25,26,27]. Recent studies in younger healthy participants [11,26,28,29,30] and stroke survivors [25] have reported high accuracies (96% to 100%) in walking step count using this approach. Moreover, shoes are ubiquitous [26] and unobtrusive [11,18,25,31,32] and, insoles are generally lightweight, thin and convenient to use [11,33]. For step count, the number of pressure sensors in insole models reported in the literature varies from one to forty-eight [11,25,26,28,29,30]. However, there appears to be no consensus for the optimal number of pressure sensors and their positioning. In most studies, pressure sensors have been positioned under heel [11,25,26,29] and under the metatarsal heads and the great toe [11,25,26]. Experiments have been performed in different environments, such as in laboratory [25,29], indoors [26] and in the community [11,28]. Data collection has been over a short time period (two minutes) [25], over a short distance (16 or 720 m) [26,28], or over a limited pre-defined number of steps (50 or 100 steps set in advance) [11,29,30]. Different signals processing methods were developed across studies, using sum and average of pressure signals. These methods yielded accuracies ranging from 96% to 100%. Although, some studies have reported nearly 100% accuracy for step counting, other studies reported errors from 1% to 4% over a short data collection period (sets of 50 and 100 steps, two minutes) [25,29,30]. Even though they may appear small, these measurement errors could lead to marked under- or over-estimation of the user’s performed steps over a whole day and impact on interpretation of results in comparison with public health recommendations regarding the number of steps per day. Very high step count accuracy is clearly needed.

To address this step count accuracy problem, we propose in the present study (1) to equip a customer-based insole integrating five force sensitive resistors (FSR) to measure foot pressure, and (2) to use a novel method of measuring and recording walking step count. The novel method of step counting was based on the cumulative sum of pressure data during the stance phase of the gait cycle. It was felt that this method would allow for only one pressure peak per step. In previous studies using the sum or average of pressure signals, there were sometimes dual peaks, which may induce an over-estimation of step counts [34,35,36]. Indeed, depending on walking speed, double peaks of pressure signals can occur during stance phase because of a possible delay of signal capture by an FSR placed under the heel and another one under the toe. These dual peaks can generate bias of counting when step detection is based on peak pressure. In contrast, when cumulative sum of pressure signals is computed during the stance phase, only one pressure peak is generated; hence, step counting error could be minimized. In the present study, we designed a system that used five FSRs positioned on different foot contact points to detect pressure and then, quantify steps. Our hypotheses were that (1) the proposed system would provide pressure signals for step count, and (2) cumulative sum of pressure signals would exhibit higher accuracy for step count in a healthy participant compared to calculation methods based on individual or averaged pressure signals.

## 2. Materials and Methods

### 2.1. Hardware

#### 2.1.1. System Overview

An instrumented insole was designed to be comfortable to wear and easy to operate. The instrumented insole did not cause observable interference with normal motion and activity. The overall architecture design is shown in Figure 1. The instrumented insole integrated (1) five FSRs; (2) a circuit board including a microcontroller module, Bluetooth module, micro-USB connector and analog-to-digital converter (ADC); (3) five resistances of 10 kilo-ohms each and (4) a battery module. The analog functions designed for the instrumented insole are described in Table 1.

#### 2.1.2. Force Sensitive Resistor (FSR)

FSR is a passive component that shows a drop in resistance when there is an increase in the force applied to the sensing area. In this study, the FSR 402 short model (Interlink Electronics, Inc., Camarillo, CA 93012, USA) with 0.6” sensing diameter was used. The FSRs 402 were thin (0.02”), durable and lightweight (0.25 g). Prior to the FSRs installation on the insole, we carried out a footprint using carbon paper with participant standing, to determine the locations under the foot with the highest pressure. Consequently, the five incorporated FSRs were located in different fore and rear foot contact points, namely the heel (FSRH), the heads of the first, third and fifth metatarsal bones (FSRM1, FSRM3, FSRM5, respectively), and the great toe (FSRT). Similar positions have been used in previous studies [18,25]. These locations of the FSRs allowed for differentiation of the most critical phases of human gait cycle, such as heel strike, foot flat, heel-off, toe-off and swing phase. For each step, a change in the resistance from the FSRs occurred, and then the microcontroller processed the signal.

#### 2.1.3. Circuit Board

We used a Dual-Core ESP-WROOM-32 module (ESP32, Shanghai, China) which supports a microcontroller, both WiFi and Bluetooth modules, a 12-bit ADC (eight channels), a micro-USB connector, a 256 kB in-system programmable flash memory and an 8-kB random-access memory. The microcontroller had to be paired with the tablet prior to data collection. This process enabled establishment of communication between the two devices. Data were sent to tablet with a speed of 9600 bits per second. The Bluetooth module allowed wireless transmission of data output from microcontroller to tablet for real-time display. The transmission distance was up to 40 m. Board-ESP32 was compatible with Arduino IDE and supported two power supply methods: USB and a 3.3-volt external lithium battery. The Board-ESP32 was positioned on the lateral side of the shoe where the instrumented insole was inserted.

#### 2.1.4. Resistance

The five resistances of 10 kilo-ohms each were placed at the ADC input of the microcontroller to vary impedance and regulate FSR sensitivity.

#### 2.1.5. Battery

The battery module contained a connector, a USB port (used to supply electric power to microcontroller) and a micro-USB module (used for charging a USB power bank). This portable and lightweight USB power bank (5 volts, 2 A) was placed in the user’s pocket and provided power to microcontroller during walking.

### 2.2. Software

The step count algorithms were developed using pressure data from each FSR individually and from all FSRs combined. The combined pressure signal was estimated using two calculation methods: average, and cumulative sum. An average pressure signal represented the mean of all five pressure signals for each sampling. The cumulative sum method consisted of the cumulative sum of all five pressure signals during the stance phase. The average [11] and sum [18,26,27] calculation methods have been used to quantify step counts in previous studies. The sampling frequency for all sensors was 10 Hz. Pressure data were pre-processed in Arduino IDE (version 1.8.5), sent to the tablet via Bluetooth module and stored in a csv file for off-line post processing using the Matlab software program.

In human gait, the stance phase is defined as the period during which the foot remains in contact with the ground (from heel strike to toe off for healthy people) [37]. Thus, during this phase, applied force was characterized by a peak, termed the pressure peak in this paper. The swing phase corresponds to the period immediately after toe off, when the foot is not in contact with ground and swings in the air for healthy people [38]. We anticipated no pressure detection for all FSRs during the swing phase.

Step count algorithms consisted of incrementing the step number for each pressure peak that corresponded to the period when the foot contacted the ground. To minimize bias, we set a delay of 500 milliseconds between two pressure peak readings in the step count algorithm. This value was selected based on pre-test data of different values ranging between 1 and 700 milliseconds. During pre-test, we observed no difference in step count with 500 milliseconds when walking at slow, self-selected and maximal speeds. In this context, each pressure peak value represented the calculated pressure during stance phase. Previous studies using average and sum methods used a pre-set threshold to minimize the detection bias [11,18,26,27]. During algorithm development, we pre-tested different thresholds: 25%, 50% and 75% of maximal force. These three thresholds led to a similar step number with no significant difference between the three thresholds. Our step count algorithms were therefore non-sensitive to thresholds, making them very simple to implement.

The step count algorithm using cumulative sum pressure of all the sensors (Algorithm 1) was defined as:(1)$$P_{sum}\left(t\right)=\overset{N}{\underset{i=1}{\sum }}P_{i}\left(t\right)\;$$
(2)$$P_{cum\_sum}\left(0\right)=P_{sum}\;\left(0\right)\;\;$$
(3)$$P_{cum\_sum}\left(t\right)=\overset{N}{\underset{i=1}{\sum }}P_{i}\;\left(t\right)+P_{cum_{sum}}\left(t-1\right)$$
where *Pi* was the value of the *i*-th pressure sensor and *N* was the number of pressure sensors. *P_sum_* was the sum of all five pressure signals and *P_cum_sum_* the cumulative sum of all five pressure signals.


**Algorithm 1: A step count algorithm using cumulative sum of all pressure signals.**

**1:   Input: N channels pressure P_i_ (t),**

**2:   Calculate the sum pressure P_sum_ (t);**

**3:   Calculate the cumulative sum pressure P_cum_sum_ (t);**

**4:   Initial: N_peak_ = step count = 0; A = 0; B = 0;**

**5:   For j = 1 : M do**

**6:           if P_heel_ ≥ 50% of P_i max_ then**

**7:                   A = Calculate P_cum_sum_ (t);**

**8:           else**

**9:               B = Calculate P_sum_ (t);**

**10:         end**

**11:         if (A != 0 && B != 0) then**

**12:           Step count = N_peak_ + 1 = step count + 1;**

**13:             A = 0; B = 0;**

**14:         else**

**15:             Step count = N_peak_ = step count;**

**16:         end**

**17: end**

**18: Return: step count.**


In order to identify the stance phase during a gait cycle, we added another condition to the algorithm, based on pressure variations under the heel. Indeed, during pre-testing, we observed that FSRH did not have any residual pressure value during the swing phase contrary to other sensors (placed under 1st, 3rd, 5th metatarsal heads and great toe). Consequently, FSRH was considered as a reference sensor for setting the condition of stance phase in the cumulative sum method. Thus, the stance phase began when the FSRH indicated a pressure value of more than 50% of the maximum pressure (50% of Pi max) and ended when the value decreased below this threshold. In the algorithm here-below, M represents the number of iterations; A and B are registers used as buffer memory (Algorithm 1).

### 2.3. Data Collection Procedure and Analysis

Twelve healthy adults without walking limitations aged from 21 to 35 (mean 28.2 ± 3.62) years old, weight from 53 to 90 kg (mean 67.8 ± 11.6 kg), and height from 1.6 to 1.83 m (mean 1.70 ± 0.08 m) participated in this study. Individual participants’ characteristics are presented in Table 2. The experimental procedure was approved by the Research Ethics Committee of Centre Intégré Universitaire de Santé et de Services Sociaux de la Capitale Nationale (CIUSSS-CN, Quebec, Canada). Informed consent was obtained from all participants. Participants wore a pair of shoes from which one was equipped with the instrumented insole. Additionally, two GaitUp sensors [39] were attached over both shoes, following the manufacturer’s instructions. Reference values for comparison were from the GaitUp Physiolog wearable measurement unit and from a direct visual step count. The latter was performed by a team member who observed each participant during the walking trials and counted the number of steps taken using a manual step counter. All trials were also videotaped for direct visual step count validation. This was used as the criterion measured value. To evaluate the accuracy of the instrumented insole, participants performed four 6-min walking trials, two at a self-selected comfortable speed (one in indoor and one outdoor) and two at maximal safe speed (one in indoor and one outdoor). Based on the pressure peak determination, we counted the number of steps for each trial and compared these data with the step counts provided by the GaitUp sensors and the manual counting. The numbers of steps detected by the insole were determined according to the methods described above in software section. The accuracy of the instrumented insole was determined as follows:(4)$$\mathrm{Accuracy}\;\left(\%\right)\;=\left(1-\;\frac{\left|\mathrm{Manual}\;\mathrm{counted}\;\mathrm{steps}\;\left(\mathrm{video}\right)\;-\mathrm{Detected}\;\mathrm{steps}\;\left(\mathrm{insole}\right)\right|}{\;\mathrm{Manual}\;\mathrm{counted}\;\mathrm{steps}\;\left(\mathrm{video}\right)}\right)\;\times \;100\%$$

## 3. Results

### 3.1. Designed Prototype of Instrumented Insole System and Quality of Signals

Pressure sensors were placed in a commercial insole that was flexible, lightweight, versatile and low-cost. Figure 1B illustrates the FSR positions which were similar to those in recent studies [18,25,40]. Moreover, this prototype could be customized to each participant.

The proposed system allowed analyzing individual or combined pressure signals, with clear recognition of stance and swing phases for each gait cycle. FSR signals were expected to approach zero during the swing phase and increase during the stance phase when a force was applied on the sensing area. However, as stated here above, we often observed low residual pressure values of FSRM1, FSRM3, FSRM5, FSRT during the swing phase as shown in Figure 2. The cumulative sum of pressure signals enabled to limit the amplitude of residual pressures during the swing phase (see Figure 3).

### 3.2. Accuracy of the Proposed System

As presented in Table 3, manual counted steps were 346 ± 36 and 351 ± 23 at self-selected walking speeds (1.43 ± 0.18 m/s and 1.45 ± 0.20 m/s), and 404 ± 31 and 416 ± 60 at maximal walking speeds (1.73 ± 0.08 m/s) and (1.78 ± 0.12 m/s) in indoor and outdoor, respectively. As expected, GaitUp provided average accuracy of 99.9 ± 0.2% and 99.8 ± 0.3% at self-selected walking speeds, 99.8 ± 0.2% and 99.8 ± 0.2% at maximal walking speeds in indoor and outdoor settings, respectively. Regarding the instrumented insole, results demonstrated the accuracy of the step counts obtained with the proposed calculation method, using one or the combination of the five FSRs. For each FSR, we obtained step count with average accuracies ranging from 94.8 ± 9.4% to 98.5 ± 2.1%, and from 96.6 ± 4.8% to 98.0 ± 2.3% at self-selected walking speeds, and from 98.1 ± 2.7% to 99.0 ± 0.7% and from 97.0 ± 6.2% to 99.4 + 0.7% at maximal walking speeds in indoor and outdoor, respectively. The calculation method based on the signals average allowed the detection of 95.5 ± 3.5% and 96.5 ± 3.9% of steps at self-selected walking speeds, and 98.6 ± 2.2% and 97.0 ± 3.6% at maximal walking speeds in indoor and outdoor, respectively. Step counts obtained with the cumulative sum calculation method was accurate at 99.5 ± 0.7% and 99.5 ± 0.4% at self-selected walking speeds, and 99.6 ± 0.4% and 99.6 ± 0.4% at maximal walking speeds in indoor and outdoor, respectively. Individual participants’ results are detailed in the Appendix A.

## 4. Discussion

This study described the design of a cheap instrumented insole integrating five FSRs for step count and presented preliminary results regarding the accuracy of step counts with the proposed system. The estimate of step counts varied according to the calculation methods. The present study showed that the cumulative sum may be an appropriate method to accurately determine the step count. Indeed, for each gait step, all the pressure data provided one pressure peak based on this method. This was an advantage because signal bias was reduced, thus increasing the accuracy of step counting. This result suggests that it is possible to estimate step counts with a high level of accuracy in healthy subjects walking at different speeds, short duration tasks, with this easy to use and cheap system and calculation algorithms. This is promising since cheap and easy to use devices with high accuracy are welcome in clinical settings for daily monitoring of physical activity.

We placed FSRs on the same positions (heel, 1st, 3rd, 5th metatarsal heads, great toe) as in Fulk et al. [25]. The choice of these positions was based on commonly reported plantar pressure in the literature. Previous studies demonstrated that rearfoot, midfoot and forefoot were successively in contact with ground during normal walking [41,42,43,44,45]. The areas with highest plantar pressure were generally heel, metatarsal and great toe [36,42,46]. In some studies, pressure sensors were positioned under heel [11,25,26,29], under heel, forefoot and great toe [11,26] for step counting. Therefore, in our study, the five selected positions for these five FSR appeared to be adequate.

Both walking speeds have been used for step counting in previous studies [20,28,29]. Similarly to Fulk et al. [25], participants in our study walked at self-selected and maximal speeds. A recent study has reported undercounting of steps (>50%) at a slow walking speed (<0.8 m/s) using pedometers and accelerometers [20]. However, other studies demonstrated 99% and 100% of accuracy at slow walking speed for step counting using sensors embedded in an insole [28,29]. Commonly, people walk at self-selected speed; therefore, it is important to ensure step counting accuracy at this walking speed. In the present study, the accuracy of the insole for step counting was high at both maximal and self-selected comfortable speeds, with cumulative sum method yielding higher accuracy than individual FSRs, and average method. These results are comparable to the results of previous studies, which observed a similar accuracy at both spontaneous and maximal walking speeds [25,28,29]. Moreover, as in some other studies [4,25,26,28,29], the accuracy of the insole for step counting did not vary according to settings (indoor or outdoor). However, step counting based on individual FSRs or signal average method showed more between-subject variability (SD between 0.7%–9.4%; and between 2.2%–3.9%, respectively) than cumulative sum method (SD ≤ 0.7%).

In comparison with the literature, test duration was longer in the present study than in most of previous studies [11,25,26,29,30], except for Bakhteri et al. [28]. Using the cumulative sum calculation method, the accuracy of step counts with the new-instrumented insole was so high as in Fulk et al. (2012) [25], where test duration was three times shorter than in our study. It might be thought that increasing the test duration did not affect accuracy level of the step count method. However, further studies using data collection of a long duration (for example, one day) are required to confirm this and to measure the pressure drift of the FSRs over a long time period. Indeed, one study reported a 6% drift after five hours of usage and a 13% drift in sensor readings after 36 h [47]. The short time of test in the present study did not allow for assessment of FSR drift.

Accuracy of individual FSRs in detecting gait step was variable (from 96.6 ± 4.8% to 99.4 ± 0.7%), with success rates sometimes lower than predicted. This was due to the unexpected residual pressure observed during the swing phase. Indeed, the presence of this residual pressure generated potential bias in the recognition of pressure peak, on which the step counting was based. For example, as shown in Figure 2, double peaks of high and low amplitudes were sometimes observed within a single gait cycle. In addition, for individual FSR, accuracy of step counts varied between participants according to walking speeds and settings. This observation may be due to the walking patterns of participants. However, accuracy of step counts computed from the FSR_H_ individual pressure-signal appeared to be less variable across walking speeds and settings. This may explain the preference for heel zone when only one FSR is used to determine step counts, as illustrated in a study by Bakhteri et al. [28]. Although one study has shown that there was no difference of vertical ground reaction force across four foot types (normal foot, pes valgus, pes cavus and hallux valgus) [48], foot type can affect pressure signals and consequently detection of steps. Further studies should evaluate accuracy for step counts considering foot types.

The step detection algorithm based on signal average led to a good detection of steps with 96.5 ± 3.9% and 95.5 ± 3.5% of accuracy at self-selected and maximal speeds, respectively. Similar values were reported in the literature with studies demonstrating step counting accuracies from 96% to 100% [11,25,26,28,29,30]. Moreover, in this study, the accuracy of step counting was more than 99% using cumulative sum without pre-set thresholds, which is an advantage compared to previous studies where the use of thresholds was necessary [11,26,28]. It is worth noting that most of the previous studies based the step count on pressure signal variation, for which a pre-set threshold is required [25,26,30,33]. In contrast, the step count method based on peak detection does not require one to set a threshold.

It has been shown that number of sensors being read at the same time affects battery life [47]. Recently, a study reported that one insole integrating three pressure sensors, one accelerometer, one gyroscope and one Bluetooth module had a time-operation of 5–6 h [49]. In contrast, another study showed that an insole incorporating nine pressure sensors and a Bluetooth module can be sustained approximatively 4 h [33]. We have tested the accuracy of individual FSRs and observed similar results as in Piau et al. [29]. Thus, the high accuracy of step counting for individual FSR might contribute to possible solution for preserving battery life. Moreover, the high accuracy of step counting for each FSR placed under different plantar areas could facilitate the adaptation of instrumented insole in different populations with various walking patterns. Therefore, further studies should examine the accuracy of different combination of FSRs to identify the minimum number of FSRs required to have valid and accurate step counts.

Some studies demonstrated that to improve usability ratings of activity monitors, many parameters had to be considered including location and comfort [20]. For instance, waist-worn activity monitors were least comfortable while the wrist-worn monitors were most comfortable [20]. When sensors and circuit board are adequately miniaturized to be comfortable and adapted for daily use, instrumented insoles may become ones of the most comfortable and thus frequently used activity monitors. Some previous studies have tested instrumented insole prototypes where the circuit board was incorporated into the insole [11,26,29]. However, similar to other studies [25,28], the circuit board (battery, microcontroller) was not integrated into our insole prototype due to module size. The proposed instrumented insole was acceptable for a short test duration, but could be uncomfortable and cumbersome for a long time use (e.g., one day). In addition, users’ walking pattern may be affected when all sensing elements are not integrated into the insole. Thus, future research should incorporate the circuit board into the insole, so as to be user-friendly and comfortable.

## 5. Conclusions

Our results indicated that instrumented insoles using force-sensitive resistors can be used to accurately quantify the number of steps taken during walking. The detection algorithm based on signals cumulative sum allows high step detection accuracy.

## Figures and Tables

**Figure 1 sensors-19-00984-f001:**
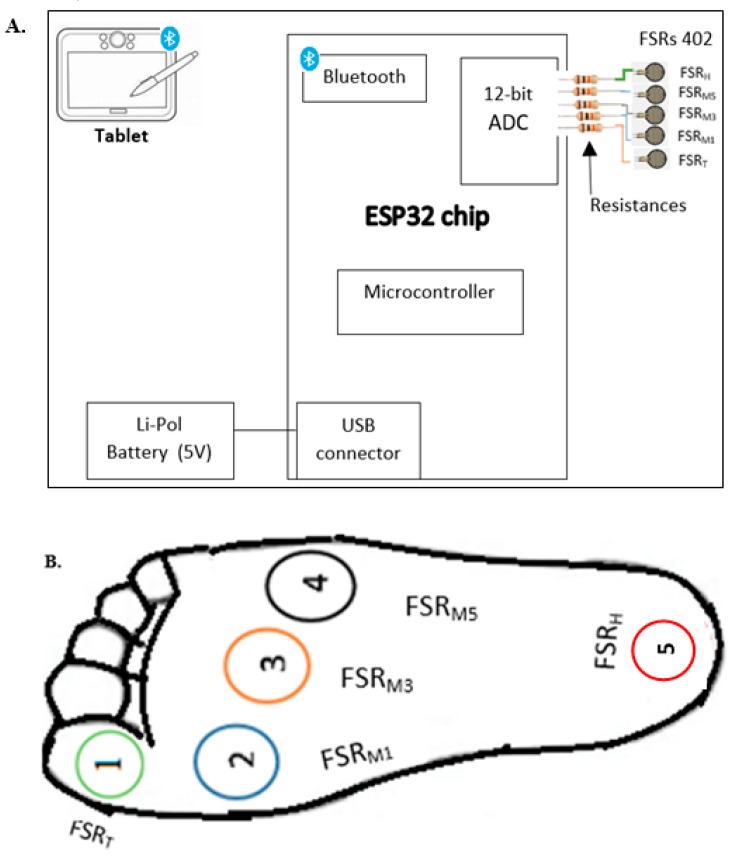
(**A**). Architecture design of instrumented insole; (**B**). Location of FSRs.

**Figure 2 sensors-19-00984-f002:**
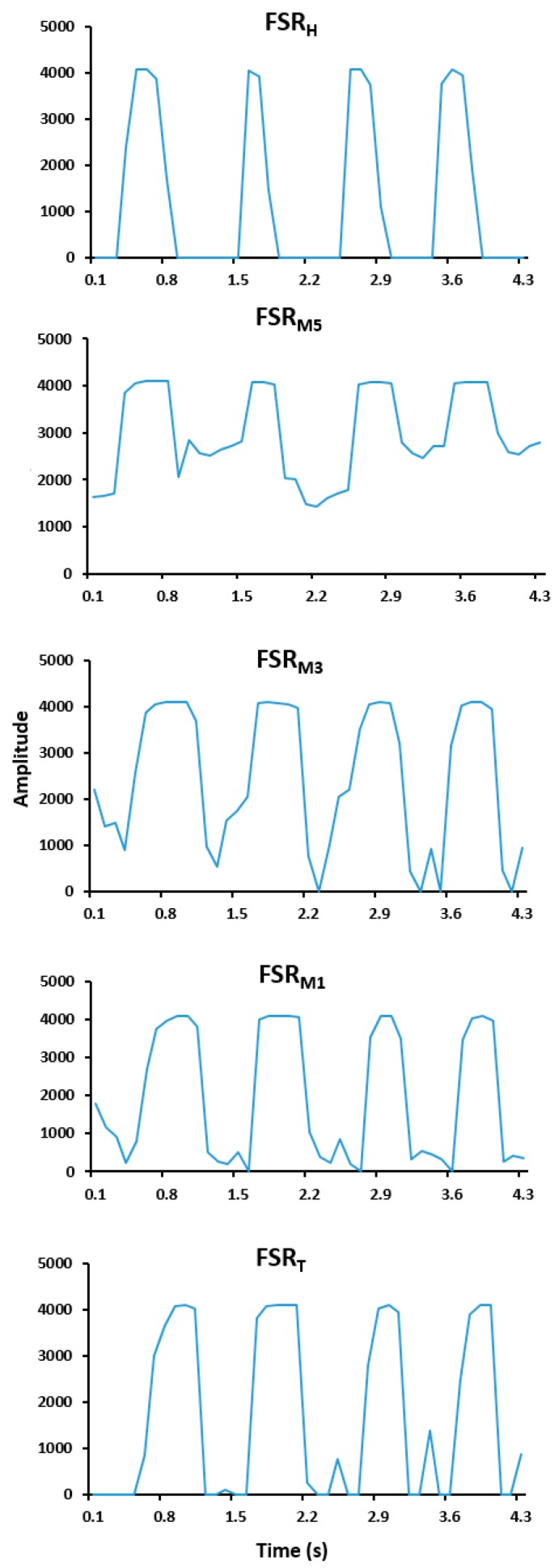
Examples of pressure signals of individual Force Sensitive Resistor (FSR) placed under the heel (FSR_H_), the heads of the first (FSR_M1_), third (FSR_M3_) and fifth (FSR_M5_) metatarsal bones (FSR_M1_, FSR_M3_, FSR_M5_, respectively), and the great toe (FSR_T_) for one participant.

**Figure 3 sensors-19-00984-f003:**
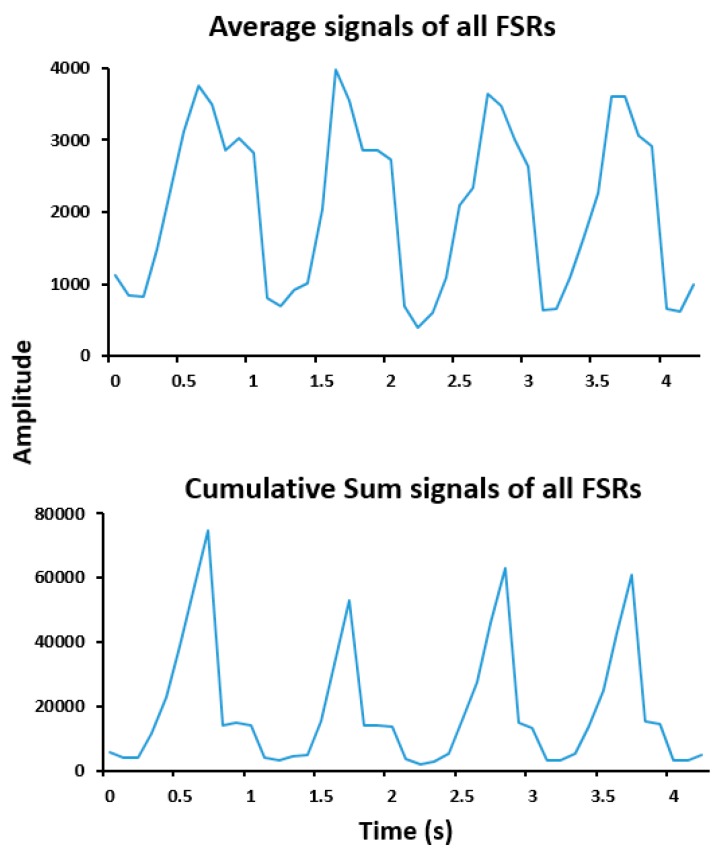
Examples of average and cumulative sum of all FSR signals obtained from one participant.

**Table 1 sensors-19-00984-t001:** Analog functions designed for the instrumented insole.

Function	Function Description	Components
Power supplying	Supply power to sensor system with a battery module of 5 V	Rechargeable lithium polymer battery
	Regulate power bank at 3.3 V	Voltage regulator of 3.3 V
Data sensing	Digitalize the resistance of FSRs	Analog-to-digital converter (8 channels) of ESP32
Data storage	Store the digital data for later processing	Memory (ESP32) and file in tablet
Data transmission	Establish wireless communication between ESP32 and tablet	Bluetooth module
Data processing	Count number of steps taken based on the digital data	Arduino IDE and Matlab Software

**Table 2 sensors-19-00984-t002:** Age, gender and anthropometric characteristics of the 12 participants.

Participants	Gender	Age (years)	Weight (kg)	Height (m)	Foot Size (cm)
1	M	29	90	1.74	28.7
2	F	25	53	1.6	24.5
3	M	27	70	1.75	24.5
4	M	28	55	1.65	24.5
5	M	26	85	1.82	27.3
6	M	31	69	1.8	27.6
7	F	31	60	1.74	26
8	F	26	62	1.65	23
9	M	35	68	1.66	26.3
10	F	29	76	1.61	25.2
11	F	21	56	1.62	23
12	M	31	70	1.83	28
F: female; M: male					

**Table 3 sensors-19-00984-t003:** Accuracies for step count using instrumented insole, compared with Gait Up and manual counting.

	Accuracy (%) ^a^, mean ± SD (n = 12)								
Settings	Walking Speed (m/s) ^b^	Individual FSR					Combined FSRs		GaitUp
		FSR_H_	FSR_M5_	FSR_M3_	FSR_M1_	FSR_T_	Average	Cumulative sum	
**Indoor**	**Self-selected speed** (1.43 ± 0.18)	98.5 ± 2.1%	98.1 ± 2.5%	97.0 ± 3.5%	94.8 ± 9.4%	96.7 ± 3.1%	95.5 ± 3.5%	99.5 ± 0.7%	99.9 ± 0.2%
	**Maximal speed** (1.73 ± 0.08)	98.4 ± 1.2%	99.0 ± 0.7%	98.1 ± 2.7%	98.6 ± 1.2%	98.6 ± 1.3%	98.6 ± 2.2%	99.6 ± 0.4%	99.8 ± 0.2%
**Outdoor**	**Self-selected speed** (1.45 ± 0.20)	98.0 ± 2.3%	96.7 ± 5.1%	97.4 ± 2.3%	96.7 ± 5.6%	96.6 ± 4.8%	96.5 ± 3.9%	99.5 ± 0.4%	99.8 ± 0.3%
	**Maximal speed** (1.78 ± 0.12)	99.4 ± 1.2%	97.0 ± 6.2%	99.0 ± 1.1%	99.0 ± 0.9%	99.4 ± 0.7%	97.0 ± 3.6%	99.6 ± 0.4%	99.8 ± 0.2%

a-Reference values provided by manual counting ; b-Walking speeds were measured with GaitUp.

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
