# Peer review of "Design and Accuracy of an Instrumented Insole Using Pressure Sensors for Step Count"

_sensors, 2019, doi:10.3390/s19050984_

Round 1
Reviewer 1 Report
Authors presented an instrumented insole with 5 FSR sensors for step counting using an algorithm based on the cumulative sum between sensors. At first, insole developments using FSR sensors were already employed in many works on the literature using very similar configurations and methods. Thus, there is no novelty in the sensor design. In addition, since the authors stated on the introduction that algorithms for step count already reached accuracies up to 100% with healthy patients and stroke survivors, the developed algorithm (evaluated only with one healthy person) seems an incremental contribution. Therefore, I cannot recommend this paper for publication in MDPI Sensors, but I encourage the authors to submit in another journal.
Author Response
We thank all reviewers for their contributive comments. Attached is a Word file with our response to reviewer 1 comment.

Reviewer 2 Report
The paper proposes a system and a related algorithm for accurate step detection using pressure-based insoles.
The paper is interesting although the authors should better highlight the real innovation of their system wrt the current state-of-the-art. While the work, in reviewer’s opinion, still need efforts to improve to a sufficient quality, the potential value is good, so in the following some comments to address:
- an actual picture of the prototype device would be appreciated
- please check algorithm A on page 5. What is M in the algorithm? The instruction to increment (line 11) the number of steps is not very clear: it seems is executed at every iteration of the for loop. Is that correct? Please clarify.
- in the evaluation section, please provide non only the accuracy metric, but also precision and recall.
- It would be interesting to know if the authors just consider the number of detected steps vs manually counted steps for obtaining the accuracy. They should indeed make sure the algorithm detects the steps where they actually are in the signal. This is related to the previous comment. In other words, to exemplify, if in a given signal there are 100 manually counted steps, and the number of steps detected by the algorithm is 100, but 20 of them are detected in wrong points in the signal, then the actual “quality” of the algorithm is way far from 100%.
- It was a good idea to present recognition results of individual sensors and of all of them altogether. Could the authors also include the evaluation of the FSR set of heel, metatarsal and great toe? It seems very relevant too.
- On the main concern with this paper is actually the weak experimental phase. Since just one subject (young healthy adult) is really not enough to have a general idea of the quality of the proposed approach in practice. The authors should carry out more experiments with a bigger and diversified subjects set.
- Finally a couple of reference suggestion:
- In the related work, the authors could consider the following reference as it could be useful for future development of the proposed work: Mei Z, Ivanov K, Zhao G, Li H, Wang L. An explorative investigation of functional differences in plantar center of pressure of four foot types using sample entropy method. Med Biol Eng Comput. 2016;55(4):537-548
- A possible reference to look at for ideas on effective embedded implementation of the proposed system could be E. Seto, M. Eladio, A.Y. Yang, P. Yan, R. Gravina, I. Lin, C. Wang, M. Roy, V. Shia, R. Bajcsy, Opportunistic Strategies for Lightweight Signal Processing for BSN. 1st International Workshop on SigProcessing, ACM PETRA 2010, Samos, Greece, 2010.
Author Response
We thank all reviewers for their contributive comments. Attached is a Word file with our point by point responses to reviewer 2 comments.
We appreciate the reviewing process and are grateful for the contributive comments of reviewers. Based on their comments we have made a number of changes which we believe have substantially improved the manuscript.

Round 2
Reviewer 1 Report
Authors addressed my concerns and clarified the paper novelty and contributions.
Reviewer 2 Report
The authors have improved the original quality of the manuscript and addressed carefully all the reviewer's comments. the paper therefore deems to be accepted.